# Topical Application of *Antrodia cinnamomea* Ointment in Diabetic Wound Healing

**DOI:** 10.3390/life12040507

**Published:** 2022-03-30

**Authors:** Ruey-Chih Su, Jyh-Gang Leu, Yuan-Hsin Chen, Chao-Yi Chen, Yi-Feng Yang, Chih-Cheng Yen, Shiu-Huey Chou, Yao-Jen Liang

**Affiliations:** 1Department of Life Science, Fu-Jen Catholic University, New Taipei City 242062, Taiwan; 054249@mail.fju.edu.tw (R.-C.S.); 407068019@mail.fju.edu.tw (Y.-H.C.); 047751@mail.fju.edu.tw (S.-H.C.); 2Graduate Institute of Applied Science and Engineering, Fu-Jen Catholic University, New Taipei City 242062, Taiwan; 401068029@mail.fju.edu.tw (C.-Y.C.); 403068073@mail.fju.edu.tw (Y.-F.Y.); 409068052@mail.fju.edu.tw (C.-C.Y.); 3School of Medicine, Fu-Jen Catholic University, New Taipei City 242062, Taiwan; M004224@ms.skh.org.tw; 4Division of Nephrology, Department of Internal Medicine, Shin Kong Wu Ho-Su Memorial Hospital, Taipei 111, Taiwan

**Keywords:** *Antrodia cinnamomea*, diabetes, ointment, wound healing, angiogenesis

## Abstract

The number of diagnosed diabetic patients is increasing worldwide. Many people with diabetes develop wounds that are slow to, or never, heal, which can lead to serious health issues. Diabetes causes long-term excessive blood glucose buildup in human body, which leads to an over-reactive inflammatory response and excessive oxidative stress. As a result, varied wound healing effects were observed according to different circumstances and stage of healing. We used two diabetic wound animal models to analyze the wound healing effect of *Antrodia cinnamomea* ointment in either topical application and/or oral administration, and explored its mechanism by Western blot analysis. The results showed that topical *Antrodia cinnamomea* treatment can significantly promote wound healing. The increased expressions of angiopoietin 1 and angiopoietin 2 protein and reduction of CD68 expression were found around wound area. Simultaneous treatment of oral and topical *Antrodia cinnamomea* ointment did not show an accelerated healing effect in our animal model. This study is the first report to demonstrate the effect of topical application of *Antrodia cinnamomea* ointment on diabetic wounds healing, and its relationship with angiogenesis. This may also open a new field for future development and application of Taiwan *Antrodia cinnamomea*.

## 1. Introduction

If diabetes is not properly controlled, it may lead to serious complications. These complications may also cause foot wounds that are unable to heal and develop into necrosis. In developed nations, diabetes is the leading cause of age-unrelated blindness, non-trauma-related amputations, and dialysis. Studies indicate that even with careful management of diabetes, 15% patients still suffer from wounds that are unable to heal at normal rates [1].

In diabetic patients, T-cell overreaction can result in apoptosis and slow healing. Diabetes can also reduce both macrophage activity and lymphatic vessels near wounds, which slow healing process. Due to the slow wound-healing, necrosis may occur as the worst outcome, along with oxidative stress to reduce tissue cell differentiation and activity [2]. As a result, it may be difficult for adjacent cells to enter the wound area and participate in repair. Diabetes is also found associated with narrowing and constriction of vasculature, which resulted in reduced blood oxygen. As a consequence, nerve ends lose their perceptual sensitivity, and reduced immune responses [3]. Oxidative stress during fibroplasias also causes fibroblasts to lose their functionality, resulting in diminished differentiation and migration capabilities, which compromises the capacity of these cells to enter wound sites and create appropriate collagen matrices [4].

The high glucose conditions worsen inflammation and oxidative stress in wound area. All of these factors combined make diabetic patients’ wounds encounter rotating inflammation and fibroplasia, which lead to ulcers or even amputations. Furthermore, diabetes can also affect DNA and many proteins, such as angiopoietin, which influences angiogenesis [5]. Impeded angiogenesis during wound-healing processes, due to impaired cellular signal transduction, results in disruption of stable new vasculature formation [6]. Finally, the diabetic patients’ wounds experience abnormal collagen homeostasis during tissue remodeling, causing fibroblast dysfunction with concomitant reduction of myofibroblasts differentiation, which then lead to diminished wound contraction and decreased formation of surface tissue [7].

*Antrodia cinnamomea* is a native Taiwanese basidiomycete with porous hymenium [8] that grows only in the inner cavity of a decayed tree trunk of *Cinnamomum kanehirai Hay* (Lauraceae). Empirical studies demonstrate that *Antrodia cinnamomea* exhibits several useful properties, including anti-cancer effects that involve inhibition of cancer cell metastasis and proliferation, protection of internal organs [9], anti-oxidation [10], and anti-inflammatory effects [11]. With regard to diabetes efficacy, *Antrodia cinnamomea* reduces blood glucose, including aberrantly high blood glucose levels [12], and it is currently available on the Taiwanese market in *Antrodia cinnamomea* beverages, which are sold as a health supplement product.

Erythropoietin (EPO) is a haemopoietic factor that is used to adjust production and differentiation of red blood cell (RBC) precursor cells [13]. EPO relies on its specific erythropoietin receptor (EPOR) to modulate blood production processes [14]. Once EPO and EPOR combine, they activate different target genes, which results in an inhibition of apoptosis [15]. EPO has been shown to stimulate vascular growth and is also effective in repairing epithelial tissue [16]. Results from diabetic mouse models indicate that EPO at the small wound site can accelerate mature capillary formation and reduce inflammation through increased epithelialization [17]. Similar results have also been obtained in rat models of diabetes for large wounds repairing [18]. The peroxisome proliferator-activated receptor superfamily (PPARs) comprises a class of nuclear receptors that help regulate multiple cellular pathways [19]. The anti-inflammatory effect of PPARδ had been shown in several previous studies [20,21]. Another study, PPARδ agonists also significantly inhibited high glucose (25 mM)-induced IL-6 and TNF-α production, NF-κB translocation, and apoptosis [20].

Prior studies reported that *Antrodia cinnamomea* is efficacious for healing tumors and aids in new vascular growth [22]. To evaluate the effects of *Antrodia cinnamomea* ointment treatments in diabetic wound healing, we investigate the efficacy of ointment and orally administered *Antrodia cinnamomea* to treat linear and large square wounds in diabetes animal models.

## 2. Materials and Methods

### 2.1. Culture of Antrodia cinnamomea Mycelia

The purchased wild *Antrodia cinnamomea* fruiting bodies were identified before experiment based on morphological features compared with a standard *Antrodia cinnamomea* strain from the Food Industry Research and Development Institute, Hsinchu, Taiwan. They were initially sterilized using 70% ethanol and antibiotics broth. After several times of washing with sterilized distilled water, small pieces of fruiting bodies were put into Eppendorf tubes with antibiotics broth and vortex. The supernatant was then transferred to the solidified Malt extract-based culture medium (MEA). Mycelia colony germinated from single spore was selected for subculture (Figure 1). The malt extract broth (MEB) was prepared for liquid suspension culture. The MEA plates or MEB suspension culture inoculated with *Antrodia cinnamomea* mycelia were kept in an incubator at 28 °C in the dark. In addition, the taxonomic identity of cultured mycelia was confirmed by isolation of genomic DNA and subjected to sequencing using ITS as marker sequence. High-performance liquid chromatography (HPLC) method [23] was also applied to measure the triterpenoids contents of *Antrodia cinnamomea* mycelia.

### 2.2. Animals

The BALB/c mice and Wistar rats (4 weeks old) were obtained from National Laboratory Animal Center, Taipei, Taiwan and the BioLASCO Taiwan Co., Ltd., Yilan, Taiwan, respectively. The BALB/c mice and Wistar rats were kept in polycarbonate cages and housed in well aerated rooms with a 12-h light/12-h dark cycle at 25 ± 2 °C, fed with standard rodent diet and water *ad libitum* in Fu Jen Laboratory Animal Center. All procedures involving animals care in this study were carried out in accordance with the recommendations set forth by the University Committee IACUC in Fu Jen Laboratory Animal Center (FJU A10526). Since this study also tested whether the effect of simultaneous oral administration of *Antrodia cinnamomea* would be better than the wound healing effect of ointment alone, the rat cutaneous wound model was selected because the volume of the solution for tube feeding *Antrodia cinnamomea* was required to be larger. It was also observed that the wound healing effect of *Antrodia cinnamomea* in different species of animals would be different. Diabetic Mice or rats only treated vehicle ointment topically as **Vehicle DM group**. Rats in **DMC group** did not receive either oral or topical treatment application. Rats treated with both topical and oral *Antrodia cinnamomea* treatment as **ACYY group**, and those accepted topical *Antrodia cinnamomea* treatment alone as **ACYN group**. **Vehicle group** was normal rats that fed with water and topical vehicle treatment.

### 2.3. Induction of Diabetic Animal

Diabetes was induced by giving intraperitoneal injections of Streptozotocin (STZ, 200 mg/kg body weight in mice; 75 mg/kg body weight in rats) [24] in BALB/c mice and Wistar rats. After one week, those with blood glucose levels higher than 200 mg/dl were judged to be diabetic. The blood sample was obtained by lancet (facial vein sampling in mice and tail vein sampling in rats) and checked by glucose oxidase method using a pre-calibrated Glucometer (TD-4227 Glucometer, Fora Care Taiwan, Taipei, Taiwan). The STZ-induced diabetic mouse and rat were then surgically wounded in the back area (1 cm linear wound in mice; 1 cm^2^ square wound in rats), which penetrate the epidermis to the fascia. They were treated by daily application of various ointments in different groups throughout the study, including the vehicle group and *Antrodia cinnamomea* group. The group of oral administration of *Antrodia cinnamomea* filtrate was applied in 2 mg/kg (total 0.5 mL in PBS) per day by oral tube feeding. When the experimental time point was reached, the mice or rats were euthanasia sacrificed to obtain skin tissue samples.

### 2.4. Ointment Preparation

The *Antrodia cinnamomea* ointment was formulated by vendors to facilitate wound smear, and completely cover the wound. Firstly, the *Antrodia cinnamomea* medicament powder was dissolved with 70% alcohol and added ddH2O up to 45 mL. Followed by adding of glycerin (Glycerin, First Chemical, Taipei, Taiwan) and emulsifier (Creagel, First Chemical, Taipei, Taiwan) in series. Rapidly stir the mix solution until solidification. The final concentration of drug paste is 1 mg/g. After applying the ointment, we attached a transparent bandage to the outside of the wound to allow the ointment to be completely absorbed.

### 2.5. Measurement of Wound Area

We use the Dino Capture photomicrography lenses (Dino-Lite; Hsinchu, Taiwan) with fixed magnification observation to record, and subject to measurement and calculation of wound size using manufacturer’s software (details on https://www.dinolite.us/en/dinocapture/ (accessed on 5 May 2021)). To quantify the healing effect of various treatment groups, the measured wound area of each animal on different days after treatment were divided by that of the first day, and the results were compared between them.

Histological procedure and Semi-Quantitative evaluation of histopathological observation are described as before and according to the published reference on Veterinarni Medicina [25] with some modification, a Five-phase scoring system was used in this study (Table 1):

### 2.6. Immunocytochemical Stain

The paraffin-embedded blocks were cut into 5-mm-thick sections to perform immunohistochemical staining using the Ventana BenchMark XT automated stainer (Ventana, Tucson, AZ, USA). Staining for the primary antibodies of Cytokeratin (Ready to use, Ventana, Tucson, AZ, USA) and Myeloperoxidase (MPO) (Ready to use, Ventana, Tucson, AZ, USA) were carried out. In each run of staining, the phosphate buffered solution was included as the negative control, and those samples that strongly express markers served as the positive controls.

### 2.7. Western Blot Analysis

A small piece of tissue was put into an Eppendorf tube with a spoonful of zirconia beads (1 mm diameter), and homogenized in cell lysis buffer (PRO-PREP). Protein concentration was determined by Bio-Rad Protein Assay Kit with Bovine Serum Albumin (BSA) as a standard. Equal amount of protein from each sample was separated by SDS-PAGE using 10% polyacrylamide gel. The separated proteins were then transferred to polyvinylidene difluoride membranes (PVDF) (Immobilon-P transfer membrane, Millipore; Burlington, MA, USA). Membranes were blocked with 5% non-fat milk for total protein in 0.1% PBST (PBS, 0.01% Tween-20) for 1 h. The membranes were then incubated with primary antibodies, CD68 (H-255), Ang-1 (C-19), Ang-2 (F-1), VEGF (VG-1), RAGE(D-5), EPOR (H-194) (1:10,000 dilution, Santa Cruz Biotechnology; Santa Cruz, CA, USA), PPAR (PP-K9436-10) (1:10,000 dilution, R&D Systems; Minneapolis, MN, USA), or actin (C-4). One hour after incubation, the membrane was washed with 1X PBST. The membranes were then incubated with a secondary antibody, i.e., rabbit anti-mouse, or goat anti-rabbit, or donkey anti-goat (1:10,000 dilution, Santa Cruz Biotechnology; Santa Cruz, CA, USA) for 1 h. After incubation, specific protein bands were detected using Immobilon Western Chemiluminescent HRP Substrate (Millipore; Burlington, MA, USA) and recorded by Kodak XAR-5 films. The signal intensity was determined by densitometry using the TotalLab TL100 v2006 software, and results are shown relative to the control value which was set to be 1 [26].

### 2.8. Statistical Analysis

All data were presented as the mean ± standard error of mean (SEM). To test statistical significance, data were subjected to unpaired one-way ANOVA by the Sigma Stat 3.5 software statistical package (Systat SigmaStat V3.5.0.54 Software; San Jose, CA, USA). Differences between groups were assessed with Fisher Least Significant Difference (LSD test) as indicated. The significance level was set at *p*-value < 0.05; * *p* ≤ 0.05; ** *p* ≤ 0.01; ^#^ *p* ≤ 0.05; ^##^ *p* ≤ 0.01.

## 3. Results

### 3.1. Effects of Antrodia cinnamomea Broth Filtrate on Linear Diabetic Wound Closure

Before ointment preparation, AC extract was tested for the potential endotoxin contamination using the RAW264.7 cell culture, i.e., a mouse macrophage cell line. There was no nitrate production observed when applying AC extract compared to the RAW264.7 cells stimulated with LPS or LPS plus interferon-γ, which resulted in various amount of activation (Appendix A). Simultaneously, the RAW264.7 cell proliferation did not significantly alter after co-treatment with various doses of AC extracts, which also suggests no cytotoxic effect of AC extracts on RAW264.7 cells (Appendix A). We have excluded the possible endotoxin contamination in our AC extract preparation.

After application of *Antrodia cinnamomea* (AC) ointment, the wounds in the treatment groups clearly contracted more than that of the vehicle group from day one through day seven (71.5 ± 12.6 vs. 45.3 ± 10.8, *p* < 0.05) (Figure 2A). Compared with the vehicle group on Day 7, the linear wounds in the *Antrodia cinnamomea* filtrate group significantly contract more (Figure 2B). These results suggested that *Antrodia cinnamomea* filtrate ointment accelerate diabetic wound healing in linear wound.

The effect of AC on wound inflammation was presented by the physical appearance of the healing wounds at day 7. By use of hematoxylin-eosin staining and observed under 400× magnification, the wound site appears as an open wound with mild epithelial hyperplasia. There were also some infiltrating cells surrounding the peripheral tissues of the wound (Figure 3). The result of Masson Trichrome stain showed that collagen fibers of control group stained blue and were abundantly found in the dermis of the dorsal skin. However, the reduction of collagen fibers was prominently ameliorated in the dermis of the AC group. When stained with cytokeratin, the wound site of control group showed epithelial hyperplasia, subsided outer edges portions of keratinized epithelium, and aggregated infiltrating cells distributed between epithelial hyperplasia. Results from myeloperoxidase stain of control group indicated that there is little aggregated distribution of infiltrating leukocytes. The Semi-Quantitative evaluation of histopathological observations is summarized in Table 2.

### 3.2. The Relationship between Antrodia cinnamomea Broth Filtrate and Angiogenesis

The vascular growth factors of diabetic patients are dysfunction, therefore, we examined the effects of treatments on various vascular growth factors. Compared with the diabetes vehicle group, those treated with *Antrodia cinnamomea* filtrate gave rise to a significant increase in Ang-1 (Figure 4A) and Ang-2 (Figure 4B) in day 7 (Appendix A). In Figure 4C, VEGF protein expression did not reach the significant difference (*p* = 0.053). These data indicated that *Antrodia cinnamomea* filtrate treatment could balance the angiogenesis process in diabetic wound healing.

### 3.3. Anti-Inflammatory Effect of Antrodia cinnamomea Broth Filtrate

CD68 is a macrophage receptor. When present at wound sites, macrophages are the first immune cells to respond to inflammation. However, if the inflammation response does not cease, macrophages will continue to accumulate at the wound site. Thus, macrophages serve as a mean of monitoring inflammatory responses [27]. In Figure 4D, the *Antrodia cinnamomea* filtrate application group showed a significant decline in the CD68 inflammation factor, compared with the diabetes vehicle group.

### 3.4. Combined Oral and Topical Antrodia cinnamomea Broth Filtrate Treatment

This study also aims to clarify whether changes in wound healing and growth factors resulted in the beneficial effect on blood glucose reduction. After oral administration of the *Antrodia cinnamomea* filtrate, no decrease in blood glucose levels was observed in the diabetic mice (data not shown). We analyzed whether combined oral and topical *Antrodia cinnamomea* filtrate treatments influence the diabetic wound healing, *n* = 6 in each group. The diabetic rats treated with topical *Antrodia cinnamomea* ointment only as Vehicle DM group. ACYN represents the rat group that was treated with topical *Antrodia cinnamomea* filtrate ointment only without feeding *Antrodia cinnamomea* filtrate. ACYY represents the group that was treated with both oral and topical *Antrodia cinnamomea* filtrate. The vehicle group was diabetic rats treated with vehicle ointment. After one-way ANOVA analysis, the wound areas of ACYN and ACYY groups significantly decreased from day 7 to day 11 in Figure 5A. In addition, oral *Antrodia cinnamomea* filtrate treatments did not result in any observable enhancement of wound healing Figure 5B. These results demonstrated that *Antrodia cinnamomea* filtrate accelerated the diabetic wound healing without lowering blood glucose; and combined oral and topical use of *Antrodia cinnamomea* filtrate are not potent in wound healing model of diabetic rats.

### 3.5. The Effects of Antrodia cinnamomea Broth Filtrate on the EPOR and PPARδ Expressions in Wound Area

EPO is a haemopoietic factor which can influence production and differentiation of progenitor red blood cell and stimulate the growth of new vasculature, while it also displays positive effects for epithelial tissue repair. In the wound healing rat model, EPOR (EPO receptor) at the wound site will not only increase the formation of mature new capillaries, but also increase epithelial growth and shorten inflammation. Therefore, we examined the EPOR levels of wound site after topical treatment alone (ACYN group) or combined with oral treatment (ACYY group), and the results showed a trend of EPOR increments but no significant difference among various groups (Figure 6A) (Appendix A). These results also indicated that EPOR signaling may not be the main factor contributing to the beneficial effect of *Antrodia cinnamomea* filtrate. On the other hand, PPARδ has been reported that participates in embryonic development and bone homeostasis [28,29], and is correlated with skin wound healing, cell growth [30], and inflammation responses [31]. Our results demonstrate that topical *Antrodia cinnamomea* filtrate treatment (ACYN group) significantly increased the PPARδ expressions. When the oral and topical treatments were combined (ACYY group), PPARδ displayed more significant changes than the other groups (Figure 6B). The vehicle group was fed with water and topical vehicle treatment. Rats in DMC group did not receive oral or topical treatment application. In sum, these results suggest that the beneficial effects of *Antrodia cinnamomea* filtrate treatment may relate to the signaling of PPARδ pathway.

## 4. Discussion

Wound healing is a very complex process. For diabetic patients, exposure to long-term elevated blood glucose, exacerbated inflammatory responses, and high oxidative stress are factors that cause difficulty for wound healing [32]. Our study found that external application of the *Antrodia cinnamomea* filtrate ointment resulted in smaller wound sites compared to the control group. The *Antrodia cinnamomea* filtrate did not exhibit potential for reducing blood glucose; however, it may affect angiogenesis and inflammation.

With regard to growth factors, we found that *Antrodia cinnamomea* filtrate treatment resulted in enhanced Ang-1 and Ang-2 levels. The Ang-1 can protect against VEGF-induced vascular permeability and maintains vascular integrity [33]. The Ang-2 can stimulate endothelial cell matrix turnover, which allows the endothelial cells to migrate and differentiate to promote angiogenesis [34]. Ang-2 substantially increases when tissues are hypoxic (while Ang-1 does not vary in this way), and excessive levels of Ang-2 can cause apoptosis, resulting in unstable angiogenesis [35]. We postulate that the control group, which received only oral administration of water, had unstable angiogenesis compared to their counterparts; when the treatment groups had already passed the onset of angiogenesis, the control group may still have remained at the initial phase of wound healing. These may explain why Ang-2 but not VEGF protein significantly increased in the wound area tissue on the day 7, and how *Antrodia cinnamomea* filtrate ointment can promote wound healing.

EPO can stimulate angiogenesis and exhibits positive effects on epithelial tissue repair [16], therefore, we also investigated if EPOR is involved. However, *Antrodia cinnamomea* filtrate did not appear to be efficacious for stimulating EPOR activity. Therefore, it indicates that *Antrodia cinnamomea* filtrate does not act on EPOR pathways in stimulating wound healing. On the other hand, PPARδ was found to be related to glucose hemostasis, inflammation, cell survival, and proliferation [36,37]. Recent study has showed that, PPARδ agonists suppressed inflammation and promoted neovascularization in the corneal wound [37]. These effects were key mechanism of skin regeneration. In the present study, we found that *Antrodia cinnamomea* filtrate could activate PPARδ expressions, inferring that may contribute to wound healing. However, more studies targeting pathway downstream of PPARδ were needed.

Combined treatment with the ointment and oral administration was tested with the rationale of stimulating diabetic wound healing both internally and externally. Moreover, this dual approach has been previously examined in studies of combined administration of oral and topical ointments to treat fungal infections of feet [38]. For each combined treatment group, the wound sites tended to become smaller in size compared to the diabetes control group and those received oral administration of *Antrodia cinnamomea* filtrate without applying topical ointment.

Based on CD68 measurements, the group that received oral administration of water without concomitant application of topical ointment had significantly increased inflammation compared to their counterparts, which imply that *Antrodia cinnamomea* filtrate has anti-inflammatory efficacy. However, there was little appreciable variation in EPOR, indicating that the effects of *Antrodia cinnamomea* filtrate may not occur via EPOR pathways. Additional study of the involved mechanisms is, thus, warranted. However, in this study, we are the first to show that oral *Antrodia cinnamomea* filtrate can activate the PPARδ expressions. In conclusion, our study provides evidence that *Antrodia cinnamomea* filtrate treatment enhances wound healing. *Antrodia cinnamomea* filtrate is primarily active in vasculature formation and anti-inflammatory efficacy. In combined administration through oral and topical pathways, the efficacy was not apparently better than through either route individually. However, overall, *Antrodia cinnamomea* filtrate showed a tendency for anti-inflammatory and angiogenesis capabilities, which suggests that it may emerge as a new pharmaceutical for diabetic wound healing.

## Figures and Tables

**Figure 1 life-12-00507-f001:**
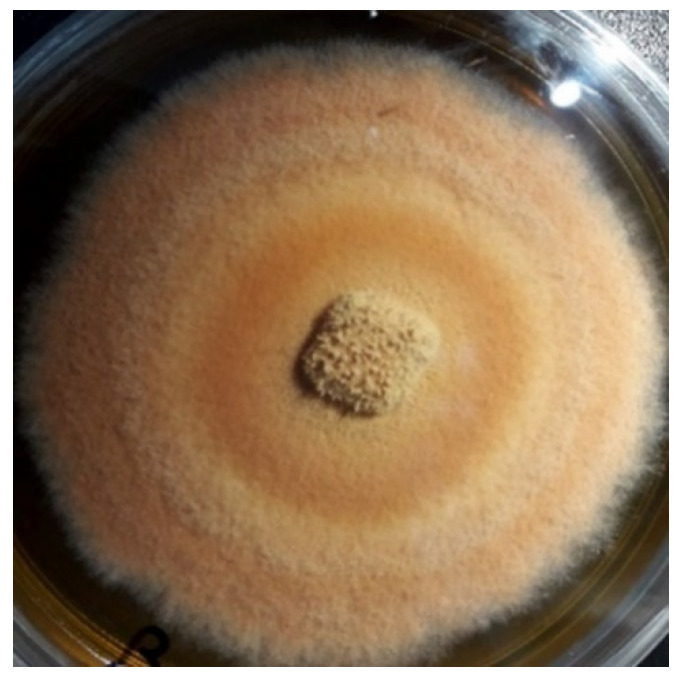
Mycelia colony germinated from single spore was selected and subcultured.

**Figure 2 life-12-00507-f002:**
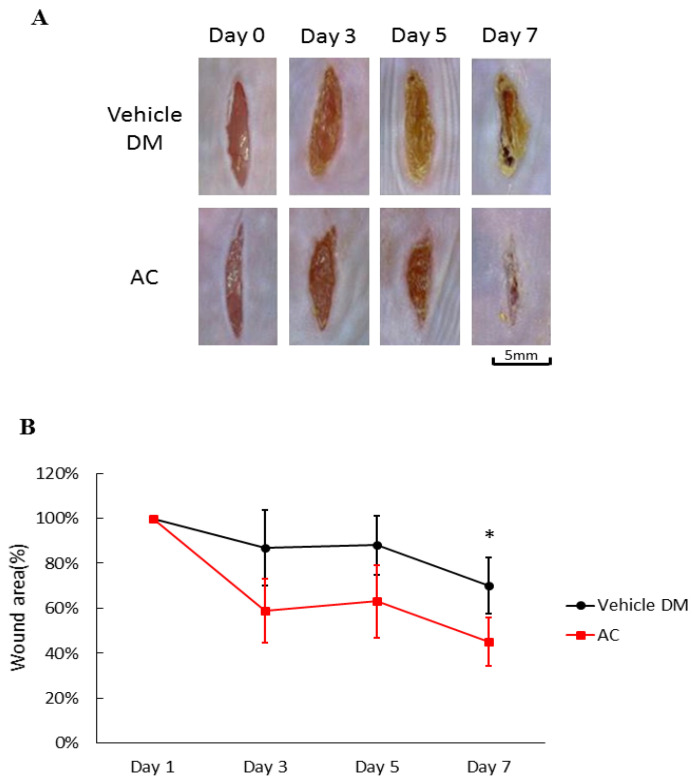
Effects of *Antrodia cinnamomea* (AC) ointment on linear cutaneous diabetic wound. (**A**) Full-thickness wounds were induced for vehicle or *Antrodia cinnamomea* treatment in diabetic mice. Images of representative mice from each group taken on post-surgery days are shown. (**B**) Mean wound area at indicated time points in each group show that healing rate in linear wound healing in diabetic mice. Each value represents the mean ± SEM; *n* = 6 for each group. * *p* < 0.05 when AC group compared to vehicle group.

**Figure 3 life-12-00507-f003:**
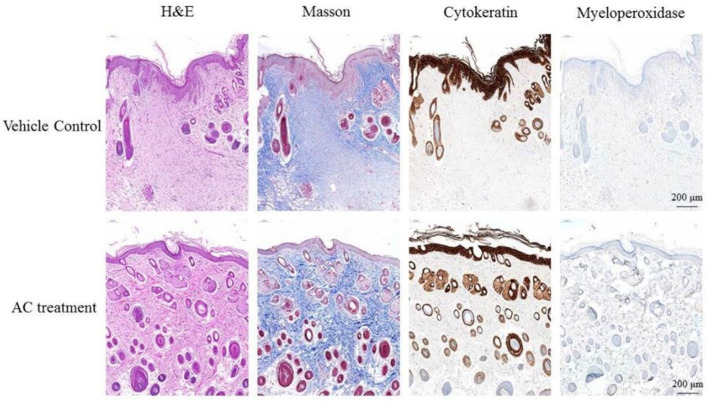
Post-wound skin tissue with specific stained in day 7. The specific stained skin sections from AC-treated diabetic mice were examined microscopically. Observation under 400× magnification.

**Figure 4 life-12-00507-f004:**
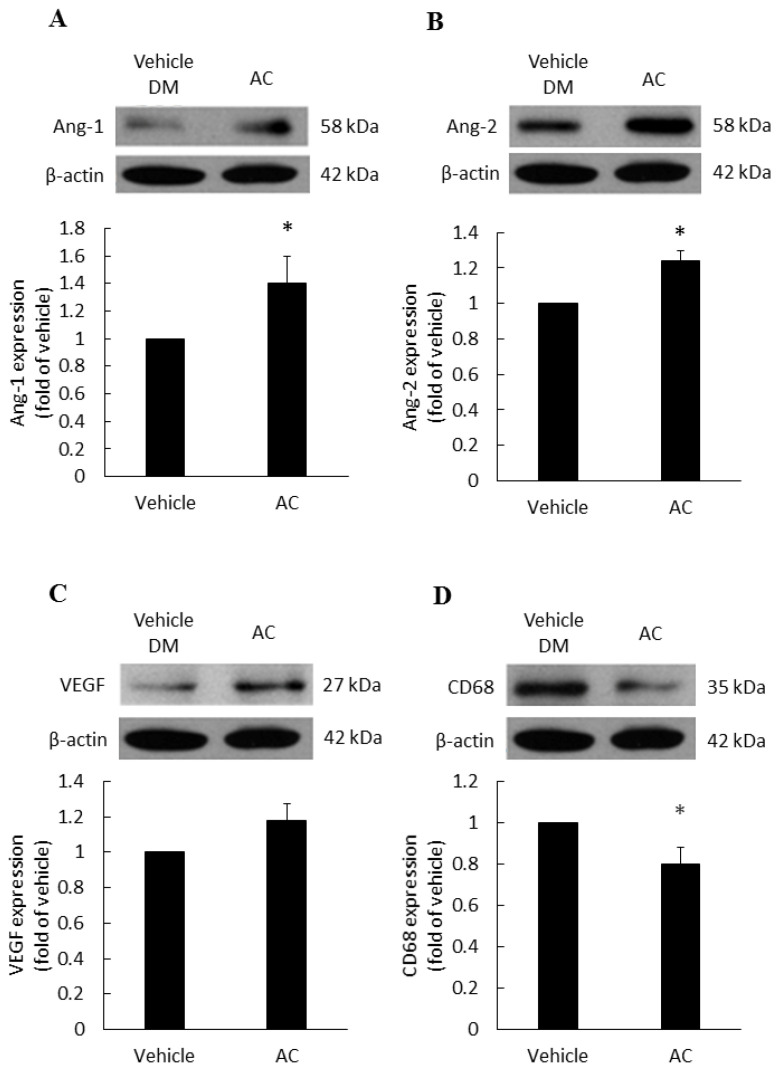
Angiogenesis and anti-inflammatory effects in diabetic wound area after topical *Antrodia cinnamomea* treatment on day 7. Angiogenesis related protein, (**A**) Angiopoietin-1 (Ang-1), (**B**) Angiopoietin-2 (Ang-2) expressions in skin tissues on Day 7 post injury. (**C**) VEGF protein expressions did not reach statistically significant. (**D**) CD68 protein expressions after the cutaneous wound in diabetic mice. Each value represents the mean ± SEM; *n* = 6 for each group. * *p* < 0.05 when AC group compared to vehicle group. (*n* = 6).

**Figure 5 life-12-00507-f005:**
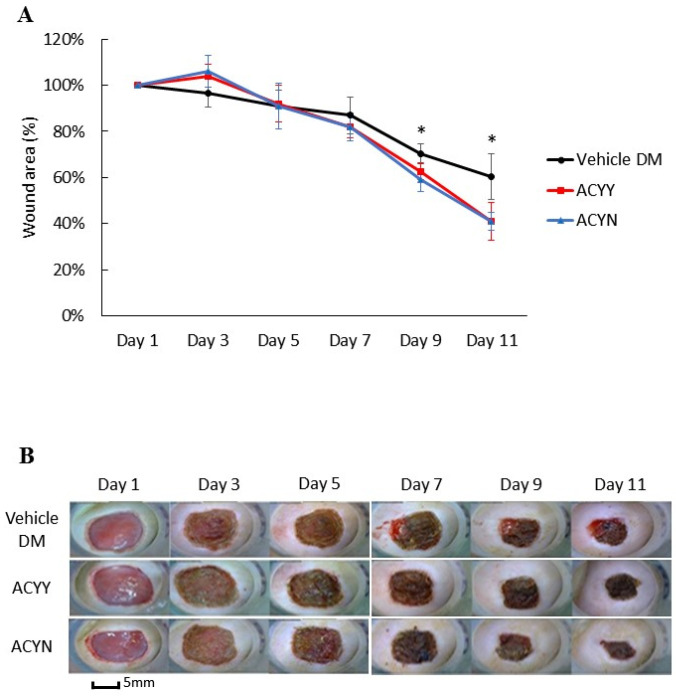
Wound healing after topical and/or oral *Antrodia cinnamomea* treatment on diabetic wounds. (**A**) Images of diabetic mice from diabetic vehicle control (Vehicle), combined topical and oral *Antrodia cinnamomea* treatment (ACYY) and topical *Antrodia cinnamomea* treatment alone (ACYN) on post-injury days are shown (**B**) Gross pictures from both combined topical and oral *Antrodia cinnamomea* treatment (ACYY) group and topical *Antrodia cinnamomea* treatment alone (ACYN) group of diabetic cutaneous wound from day 9 to day 11 in diabetic rats. Each value represents the mean ± SEM; *n* = 6 for each group. * *p* < 0.05 when treatment group compared to control group.

**Figure 6 life-12-00507-f006:**
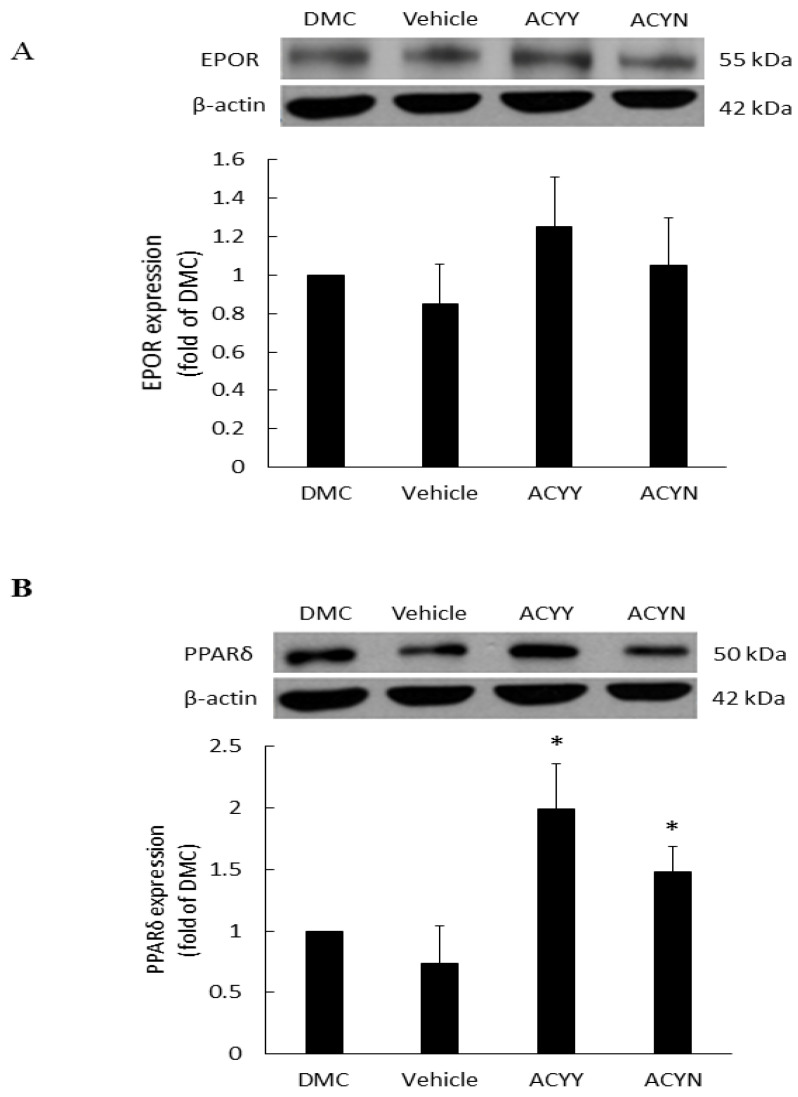
Expression of EPOR and PPARδ proteins in diabetic wound areas after *Antrodia cinnamomea* treatment. The diabetic rats without any treatment were DMC group. Vehicle group was only topically treated with vehicle ointment. (**A**) EPOR protein expressions after *Antrodia cinnamomea* treatment. (**B**) PPARδ protein expressions after combined topical and oral *Antrodia cinnamomea* treatment (ACYY) and topical *Antrodia cinnamomea* treatment alone (ACYN). Each value represents the mean ± SEM; *n* = 6 for each group. * *p* < 0.05 when treatment group compared to control group.

**Table 1 life-12-00507-t001:** Five-phase scoring system.

Grading	Epithelization	PMN	Fibroblast	Collagen
0	Thickness of cut edges	Absent	Absent	Absent
1	Migration of cells (<50%)	Mild ST	Mild-ST	Minimal-GT
2	Migration of cells (>50%)	Mild DL/GT	Mild-GT	Mild-GT
3	Bridging the excision	Moderate DL/GT	Moderate-GT	Moderate-GT
4	Keratinization	Marked DL/GT	Marked-GT	Marked-GT

PMN: polymorphonuclear leucocyte, ST: surrounding tissue, DL: demarcation line. GT: granulation tissue.

**Table 2 life-12-00507-t002:** The semi-quantitative scoring of the king in control and AC treatment.

	Control	AC	*p* Value
Epithelization	2.8 ± 0.13	3.7 ± 0.15	<0.001
PMN infiltration	2.0 ± 0.21	0.4 ± 0.16	<0.001
Fibroblast	2.0 ± 0.15	0.5 ± 0.17	<0.001
Collagen	1.9 ± 0.10	0.6 ± 0.16	<0.001

PMN = Polymorphonuclear leukocytes. Data are shown as mean ± SE.

## Data Availability

The data presented in this study are available on request from the corresponding author.

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
