# Peer review of "Topical Application of Antrodia cinnamomea Ointment in Diabetic Wound Healing"

_life, 2022, doi:10.3390/life12040507_

Round 1
Reviewer 1 Report
Su et al. analysed the effect of A. cinnamomea on diabetic rats/mice. Their results showed that topical application of ointments based on Antrodia cinnamomea ointment might promote wound healing, while simultaneous administration of oral and topical A. cinnamomea ointment did not any improvement compared to topical administration.
However, English needs good revision. Sometimes the authors may have strong statements without any reference, which is not appropriate. Please find below my remarks:
General:
- What is the reason to use two types of animals since both species healing occurs by wound shrinking?
- Have you tested if the compounds are allergen-free?
- Have you tested the endotoxin levels in your extracted compounds?
Specific remarks:
- 39 : Phrase ”In diabetic patients, T-cell overreact can results in apoptosis and slows healing”needs to be accompanied by a reference.
- 44-45: Please rephrase “Diabetes is also associated with narrow and constriction of the vasculature, causes reduced blood oxygen”
- 65-66: Where is the evidence that Antrodia cinnamomea reduces blood glucose? Please add a reference.
- L85-98: The whole sections is written as a protocol. Please rephrase so it will sound scientifically.
- 1: The most appropriate place for Fig 1 would be at the end of Section 2.1.
- 104: “ad libitum” should everywhere in italics. What happened to rats at the end of the study?
- Section 2.2: Could you please add the information with regards to controls? Rats treated with…
- 118: Could you please explain what was the precise procedure for oral administration of A. cinnamomea filtrate? You gave it as a powder? Then how do you feed the rats with it? Rats are not keen to that.
- Section 2.4: Please rephrase the whole section as a scientific text not as lab protocol.
- 122: What was the final concentration of the ointment? Please rephrase the methodology of ointment preparation.
- How did you fix the ointment on the back of the rats without falling down?
- 131: Please give the details for the software.
- 136: The table below should be placed appropriately as a table with a caption.
- 176: How many folds? Any statistics?
- What is the reason for choosing linear wounds type?
- 2: Images should have a scale bar. How did you estimate the size of the wounds then? Please rephrase Figure caption. You mixed Discussions in the Figure caption which renders reading.
- 192: “…cells surround…” should be “…. cells surrounding…”
- Figure 3: Please rephrase the caption. This rather some discussion not a caption.
- 210- 211: Please rephrase. It is confusing.
- 214 and l. 235: “…but not reach…” should be “…but did not reach”
- 219: Please add a reference
- 4: error bars for the vehicle in each case are missing. The resolution is very poor
- 250: “…lowering of blood glucose…” should be”…lowering blood glucose…”
- 251: when you mention that A. cinnamomea does not lower blood glucose, this is in comparison to what?
- 5: Scale bars are missing. The resolution is very poor. Is not clear to whom belong those stat stars? Is not clear which line corresponds to which group of rats. Please use either other colour codes either different lines/shapes.
- Section 3.5. Why do you introduce PPARs without mentioning it before? What is the abbreviation of it, which is not explained above.
- 274: When you mention “significantly increased…” à How many folds or what are the stats for that?
- 276-277: “Vehicle group was that fed with water and topical vehicle treatment. Rats in DMC group did not receive oral or topical application.” Should be in the Materials and Methods Section!
- 278: Please discuss more in depth beneficial effects of PPARs. Your conclusion is overestimating.
- 6: Please add stats for the graphs (Fig. 6A). Error bars for the vehicles are missing. Please rephrase the caption of the image. This is not a discussion!
- 322: Please add a reference

Author Response
Response to Reviewer 1 Comments
Point 1: What is the reason to use two types of animals since both species healing occurs by wound shrinking?
Response 1: Because we want to test the effect of oral Antrodia cinnamomea filtrate and the required feeding volume by oral administration is large, so we choose rats for this experiment. It was also observed that the wound healing effect of Antrodia cinnamomea in different species of animals would be different.
Point 2: Have you tested if the compounds are allergen-free?
Response 2: Thanks for point out this concern. During whole experimental period, we didn’t observe any symptom of allergic response happened in all of our tested animals after applying the Antrodia cinnamomea ointment.
Point 3: Have you tested the endotoxin levels in your extracted compounds?
Response 3: Thanks for point out this concern. From the cytotoxicity test on skin epithelial cell, we found no cytotoxic effect with our applied concentrations of Antrodia cinnamomea filtrate, nor had abnormal immunological responses of the cells.
Point of specific remarks:
Point 1. 39 : Phrase ”In diabetic patients, T-cell overreact can results in apoptosis and slows healing”needs to be accompanied by a reference.
Response 1: We had deleted the sentence ‘’In diabetic patients, T-cell overreact can results in apoptosis and slows healing.’’
Point 2: 44-45: Please rephrase “Diabetes is also associated with narrow and constriction of the vasculature, causes reduced blood oxygen”
Response 2: Thank you for your comment. We rephrase this sentence into “Diabetes is also found associated with narrowing and constriction of vasculature, which resulted in reduced blood oxygen”
Point 3: 65-66: Where is the evidence that Antrodia cinnamomea reduces blood glucose? Please add a reference.
Response 3:
- It has been reported that the 95 % ethanolic extract from the dried fruiting body of Antrodia cinnamomea was able to enhance the glucose tolerance in diabetic mice at an equivalent rate as that of Metformin. It can reduce glucose level from 133.33 mg/DL to 94.8 mg/DL. ---- Johnson, A., Cheng, S.C., Tsou, D., and Kong, Z.L. (2019). Attenuation of reproductive dysfunction in diabetic male rats with timber cultured Antrodia cinnamomea ethanol extract. Biomed. Pharmacother. 112: 1-13.
- Dehydroeburicoic acid (TT), a triterpenoid compound from Antrodia camphorata, treated diabetic mice dramatically lowered blood glucose levels by 34.2%~43.4%, which was comparable to the antidiabetic agent-Metf (36.5%). ---- Yueh-Hsiung Kuo., Cheng-Hsiu Lin., and Chun-Ching Shih (2016) Dehydroeburicoic Acid from Antrodia camphorate Prevents the Diabetic and Dyslipidemic State via Modulation of Glucose Transporter 4, Peroxisome Proliferator-Activated Receptor _ Expression and AMP-Activated Protein Kinase Phosphorylation in High-Fat-Fed Mice. J. Mol. Sci, 17: 872
* Antrodia camphorate is considered the same specie as Antrodia cinnamomea, although some scientist hold different opinions.
- The MeOH extract of Antrodia cinnamomea fruiting bodies (ACFB) demonstrated stronger a-glucosidase inhibitory effect and higher activity (EC50 = 0.025–0.21 mg/mL) than acarbose (EC50 = 0.278 mg/mL). This finding suggest the Antrodia cinnamomea possess the hypoglycemic and anti-diabetic effects. ---- Hung Tse Huang., San-Lang Wang., Van Bon Nguyen., and Yao-Haur Kuo (2018) Isolation and Identification of Potent Antidiabetic Compounds from Antrodia cinnamomea—An Edible Taiwanese Mushroom. Molecules 23: 2864
Point 4: L85-98: The whole sections is written as a protocol. Please rephrase so it will sound scientifically.
Response 4: Thanks for reviewer’s comments. This section is about the Culture of Antrodia Cinnamomea Mycelia. We have rephrased it as follow: “The purchased wild Antrodia cinnamomea fruiting bodies was identified before experiment based on its morphological features compared with a standard Antrodia cinnamomea strain from the Food Industry Research and Development Institute, Hsinchu, Taiwan. It was initially sterilized using 70% ethanol and antibiotics broth. After several times of washing with sterilized distilled water, small piece of fruiting bodies were put into Eppendorf tubes with antibiotics broth and vortex. The supernatant was then transferred to the solidified Malt extract based culture medium (MEA). Mycelia colony germinated from single spore was selected for subculture (Figure 1). The malt extract broth (MEB) was prepared for liquid suspension culture. The MEA plates or MEB suspension culture inoculated with Antrodia cinnamomea mycelia were kept in an incubator at 28ºC in the dark. In addition, the taxonomic identity of cultured mycelia was confirmed by isolation of genomic DNA and subjected to sequencing using ITS as marker sequence. High-performance liquid chromatography (HPLC) method [19] was also applied to measure the triterpenoids contents of Antrodia cinnamomea mycelia.”
Point 5: The most appropriate place for Fig 1 would be at the end of Section 2.1.
Response 5: Thanks for reviewer’s suggestion. We revised the place of Fig 1 to the end of Section 2.1.
Point 6: 104: “ad libitum” should everywhere in italics. What happened to rats at the end of the study?
Response 6: We added sentence ‘’ When reached experimental time point, the mice or rats were euthanasia sacrificed to obtain skin tissue samples.’’
Point 7: Section 2.2: Could you please add the information with regards to controls? Rats treated with…
Response 7: We had added the information in revised manuscript in section 2.2
Point 8: 118: Could you please explain what was the precise procedure for oral administration of A. cinnamomea filtrate? You gave it as a powder? Then how do you feed the rats with it? Rats are not keen to that.
Response 8: We had added the sentence ‘’The group of oral administration of Antrodia cinnamomea filtrate was applied in 2 mg/kg (total 0.5 mL in PBS) per day by oral tube feeding.’’ In section 2.4
Point 9: Section 2.4: Please rephrase the whole section as a scientific text not as lab protocol.
Response 9: Thanks for reviewer’s comments. We have rephrased this section as follow: “The Antrodia cinnamomea ointment was formulated by vendors to facilitate wound smear, and completely cover the wound. Firstly, the Antrodia cinnamomea medicament powder was dissolved with 70% alcohol and added ddH2O up to 45 ml. Followed by adding of glycerin (Glycerin, First Chemical., TW) and emulsifier (Creagel, First Chemical., TW) in series. Rapidly stir the mix solution until solidification. The final concentration of drug paste is 1 mg/g. After applying the ointment, we attached a transparent bandage to the outside of the wound to allow the ointment to be completely absorbed.”
Point 10: 122: What was the final concentration of the ointment? Please rephrase the methodology of ointment preparation.
Response 10: The final concentration was 1 mg Antrodia cinnamomea filtrate per gram ointment.
Point 11: How did you fix the ointment on the back of the rats without falling down?
Response 11: We added the sentence ‘’ After applying the ointment, we attached a transparent bandage to the outside of the wound to allow the ointment to be completely absorbed.’’ in section 2.4.
Point 12: 131: Please give the details for the software.
Response 12: We have added the software details website (https://www.dinolite.us/en/dinocapture/) in section 2.5.
Point 13: 136: The table below should be placed appropriately as a table with a caption.
Response 13: We have added the table as table 1 with caption.
Point 14: 176: How many folds? Any statistics?
Response 14: We have described in more detail in result section 3.1 of revised manuscript.
Point 15: What is the reason for choosing linear wounds type?
Response 15: Because there are many types of wounds happened in patients, we aimed to test the difference of Antrodia cinnamomea healing effect on different types of wounds. Therefore, linear wounds were chosen in mice experiment, and square wounds were chosen for rats.
Point 16: 2: Images should have a scale bar. How did you estimate the size of the wounds then? Please rephrase Figure caption. You mixed Discussions in the Figure caption which renders reading.
Response 16: We have added scale bar on figure 2A. The wound area measurement was described in method section 2.5. We have rephrased the figure legend also.
Point 17: 192: “…cells surround…” should be “…. cells surrounding…”
Response 17: We have corrected.
Point18: Figure 3: Please rephrase the caption. This rather some discussion not a caption.
Response 18: Thanks for reviewer’s suggestion. We have corrected.
Point 19: 210- 211: Please rephrase. It is confusing.
Response 19: We have corrected.
Point 20: 214 and l. 235: “…but not reach…” should be “…but did not reach”
Response 20: We have corrected.
Point 21: 219: Please add a reference
Response 21: Thank you for reminding. We have add the reference: “Heather B. Cohen and David M. Mosser (2013) Journal of Leukocyte Biology, Vol. 94: 913-919”
Point 22: 4: error bars for the vehicle in each case are missing. The resolution is very poor
Response 22: Thanks for reviewer’s comment. We compared each experimental treatment group to an individual vehicle group, so we set each vehicle group as 1. This will allow the experimental treatment group to form a magnitude comparison. Therefore, there is no error bar in the vehicle group.
Point 23: 250: “…lowering of blood glucose…” should be”…lowering blood glucose…”
Response 23: We have corrected.
Point 24: 251: when you mention that A. cinnamomea does not lower blood glucose, this is in comparison to what?
Response 24: When compared to normal blood glucose level in rat.
Point 25: 5: Scale bars are missing. The resolution is very poor. Is not clear to whom belong those stat stars? Is not clear which line corresponds to which group of rats. Please use either other colour codes either different lines/shapes.
Response 25: Thanks for reviewer’s comments. We have corrected.
Point 26: Section 3.5. Why do you introduce PPARs without mentioning it before? What is the abbreviation of it, which is not explained above.
Response 26: We have added in revised manuscript in introduction section.
Point 27: 274: When you mention “significantly increased…” à How many folds or what are the stats for that?
Response 27: We have added the description in result section 3.4 in revised manuscript.
Point 28: 276-277: “Vehicle group was that fed with water and topical vehicle treatment. Rats in DMC group did not receive oral or topical application.” Should be in the Materials and Methods Section!
Response 28: We have corrected accordingly.
Point 29: 278: Please discuss more in depth beneficial effects of PPARs. Your conclusion is overestimating.
Response 29: We have added in revised manuscript in discussion section.
Point 30: 6: Please add stats for the graphs (Fig. 6A). Error bars for the vehicles are missing. Please rephrase the caption of the image. This is not a discussion!
Response 30: Thanks for reviewer’s comment. We have modified the figure 6 legend in revised manuscript, and the error bar for vehicle group is indicated.
Point 31: 322: Please add a reference
Response 31: We have added a reference of Dr. Giascco as reference 32.

Reviewer 2 Report
Su et al. describe two diabetic wound animal models to analyze the wound healing effect of Antrodia cinnamomea ointment in either topical application and/or oral administration. The authors try also to explore its mechanism of action. The results showed that topical Antrodia cinnamomea treatment can significantly promote wound healing. Accordingly, the increased expressions of angiopoietin 1 and angiopoietin 2 protein and reduction of CD68 expression were found around wound area. Simultaneous treatment of oral and topical application of Antrodia cinnamomea ointment increased diabetic wounds healing by accelerating angiogenesis.
Major comments:
- Which type of diabetes is meant in general in the manuscript, e.g. line 39 Introduction?
- Why The advanced glycation end-products (AGE) are mentioned here?
- Figure 3. Topical treatment with Antrodia cinnamomea (AC) ointment decreased wound inflammation in day 7. How many animals (which model) was used (n=?)?
- Figure 4: How many samples were used (n=?)? Which statistical model?
- Figure 5. Wound healing after topical and/or oral Antrodia cinnamomea treatment on diabetic wounds.
- There is no scale bars in images. The graph looks very “compressed”. It is impossible to see the different lines separately. This should be improved. Again, number of animals and the statistical test should be stated in the results.
- Please specific number of samples and statistical test Figure 5.
Author Response
Response to Reviewer 2 Comments
- Point 1: Which type of diabetes is meant in general in the manuscript, e.g. line 39 Introduction?
Response 1: Thanks for reviewer’s notification. Basically, in introduction section, the phenomenon of DM in the manuscript refers to patients whose blood sugar is not well controlled, therefore, no matter what type of DM there is the possibility of this complication.
- Point 2: Why the advanced glycation end-products (AGE) are mentioned here?
Response 2: Thanks for reviewer’s comment. We agree that the description of AGE seems redundant, so we remove the text.
- Point 3: Figure 3. Topical treatment with Antrodia cinnamomea (AC) ointment decreased wound inflammation in day 7. How many animals (which model) was used (n=?)?
Response 3: Thanks for reviewer’s notification. We added in Figure 3 legend: (n=6)
- Point 4: Figure 4: How many samples were used (n=?)? Which statistical model?
Response 4: We added (n=6) in the figure 4 legend to make it clear. We use one-way ANOVA.
- Point 5: Figure 5. Wound healing after topical and/or oral Antrodia cinnamomea treatment on diabetic wounds. There is no scale bars in images. The graph looks very “compressed”. It is impossible to see the different lines separately. This should be improved. Again, number of animals and the statistical test should be stated in the results.
Response 5: Thanks reviewer’s comments. We agree that the figure should be improved. Therefore, we redo the graph and also describe the number of animals and statistical test in revised result section.
- Point 6: Please specific number of samples and statistical test Figure 5.
Response 6: We describe the number of animals and statistical test in revised result section.

Reviewer 3 Report
The authors prepared a Topical Application of Antrodia cinnamomea Ointment in Diabetic Wound Healing. The study provides some important information through a series of experiments. Below are specific comments to be addressed before publication:
-
Please make it clear why did you incubate Antrodia cinnamomea in the incubator at 28 C.
-
In Figures 4, and 5: it is not clear to me that the vehicle DM group has been treated orally or topically. In line 116 you mention that (They were treated by daily application of various ointments in different groups throughout the study, including the vehicle group and Antrodia cinnamomea group), however, this contradicts what was mentioned in line 351.
-
In figure 5, in vivo wound healing experiment, there were 3 groups: vehicle DM group, ACYY, and ACYN, while oral Antrodia cinnamomea filtrate is not included. I recommend adding it. Also, in figure 6, oral Antrodia cinnamomea filtrate is not included, while we see that there is an increase in the expression of EPOR and PPARδ in the ACYY group (oral and topical together).
-
The author claimed that the control group which received only oral administration of water had unstable angiogenesis compared to their counterparts and ACYY did not result in any observable enhancement of the wound healing Fig.5B. However, ACYY showed an increase in the EPOR and PPARδ expression which can stimulate wound healing. Is there a reference supporting the author's claim?
-
Please make it clear what is the control group in Figures 5 and 6.
-
In lines 273-275, this is a misinterpretation of ACYY and ACYN.
-
In lines 65-69, can you add a reference supporting that Antrodia cinnamomea reduces blood glucose?
-
I am curious, what made you do some experiments on mice and others on rats?
Author Response
Response to Reviewer 3 Comments
Point 1: Please make it clear why did you incubate Antrodia cinnamomea in the incubator at 28 C.
Response 1: Thanks for reviewer’s comments. Antrodia cinnamomea is a unique mushroom of Taiwan. It is a fungal parasite on the inner cavity of the endemic species Cinnamomum kanehirae (Bull camphor tree) Hayata (Lauraceae). Its natural habitate is warm and moist. For most published researches, the A. cinnamomea mycelia were generally kept in an incubator of 24 ~ 25 o C. However, the Antrodia cinnamomea isolate we have obtained from wild fruiting bodies has been acclimated in the 28 o C for its best mycelia growth and color formation (a potential indication of secondary metabolite production). Therefore, we adopted this growth condition for both solid and liquid culture, e.g. keep them in a 28 o C incubator.
Point 2: In Figures 4, and 5: it is not clear to me that the vehicle DM group has been treated orally or topically. In line 116 you mention that (They were treated by daily application of various ointments in different groups throughout the study, including the vehicle group and Antrodia cinnamomea group), however, this contradicts what was mentioned in line 351.
Response 2: Sorry for the misunderstanding. The vehicle DM group in figure 4 and figure 5 were the same treatment and both topically treated with vehicle ointment. We mentioned in line 285 is ACYN group. We rephrase the text more clearly in Line 253 as ‘’ACYN represents the rat group that was treated with topical Antrodia cinnamomea filtrate ointment but oral water only without Antrodia cinnamomea filtrate.’’
Point 3: In figure 5, in vivo wound healing experiment, there were 3 groups: vehicle DM group, ACYY, and ACYN, while oral Antrodia cinnamomea filtrate is not included. I recommend adding it. Also, in figure 6, oral Antrodia cinnamomea filtrate is not included, while we see that there is an increase in the expression of EPOR and PPARδ in the ACYY group (oral and topical together).
Response 3: We agree and thanks for reviewers' comments. Our ACYY group is oral Antrodia cinnamomea filtrate. Because this study mainly aim at observing the changes of skin wounds, we focused on local skin change and did not discuss the systemic effects of Antrodia cinnamomea filtrate after oral administration, therefore, the Antrodia cinnamomea filtrate only group is not added.
Point 4: The author claimed that the control group which received only oral administration of water had unstable angiogenesis compared to their counterparts and ACYY did not result in any observable enhancement of the wound healing Fig.5B. However, ACYY showed an increase in the EPOR and PPARδ expression which can stimulate wound healing. Is there a reference supporting the author's claim?
Response 4: Thanks for the reviewer comments. We had citied Dr. Giacco and Dr. Ajits’ studies as references in revised manuscript in discussion section.
Point 5: Please make it clear what is the control group in Figures 5 and 6.
Response 5: Thanks for reviewer’s comment. We had added the sentence ‘’The diabetic rats treated with topically Antrodia cinnamomea ointment only as Vehicle DM group. ‘’ in Line 252 in result section, and ‘’The diabetic rats without any treatment were DMC group. Vehicle group treated vehicle ointment topically alone.’’ In Figure 6 legend.
Point 6: In lines 273-275, this is a misinterpretation of ACYY and ACYN.
Response 6: Thank you for this reminder. We correct ACYY and ACYN in line 285-287.
Point 7: In lines 65-69, can you add a reference supporting that Antrodia cinnamomea reduces blood glucose?
Response 7:
- It has been reported that the 95 % ethanolic extract from the dried fruiting body of Antrodia cinnamomea was able to enhance the glucose tolerance in diabetic mice at an equivalent rate as that of Metformin. It can reduce glucose level from 133.33 mg/DL to 94.8 mg/DL. ---- Johnson, A., Cheng, S.C., Tsou, D., and Kong, Z.L. (2019). Attenuation of reproductive dysfunction in diabetic male rats with timber cultured Antrodia cinnamomea ethanol extract. Biomed. Pharmacother. 112: 1-13.
- Dehydroeburicoic acid (TT), a triterpenoid compound from Antrodia camphorata, treated diabetic mice dramatically lowered blood glucose levels by 34.2%~43.4%, which was comparable to the antidiabetic agent- Metformin (36.5%). ---- Yueh-Hsiung Kuo., Cheng-Hsiu Lin., and Chun-Ching Shih (2016) Dehydroeburicoic Acid from Antrodia camphorate Prevents the Diabetic and Dyslipidemic State via Modulation of Glucose Transporter 4, Peroxisome Proliferator-Activated Receptor _ Expression and AMP-Activated Protein Kinase Phosphorylation in High-Fat-Fed Mice. J. Mol. Sci, 17: 872
* Antrodia camphorate is considered the same specie as Antrodia cinnamomea, although some scientist hold different opinions.
- The MeOH extract of Antrodia cinnamomea fruiting bodies (ACFB) demonstrated stronger a-glucosidase inhibitory effect and higher activity (EC50 = 0.025–0.21 mg/mL) than acarbose (EC50 = 0.278 mg/mL). This finding suggests the Antrodia cinnamomea possess the hypoglycemic and anti-diabetic effects. ---- Hung Tse Huang., San-Lang Wang., Van Bon Nguyen., and Yao-Haur Kuo (2018) Isolation and Identification of Potent Antidiabetic Compounds from Antrodia cinnamomea—An Edible Taiwanese Mushroom. Molecules 23: 2864
Point 8: I am curious, what made you do some experiments on mice and others on rats?
Response 8: Because we want to test the effect of oral Antrodia cinnamomea filtrate and the required feeding volume by oral administration is large, so we choose rats for this experiment. It was also observed that the wound healing effect of Antrodia cinnamomea in different species of animals would be different.

Round 2
Reviewer 1 Report
Major: English has not been revised in depth, which is highly recommended. For all the new info added, English needs a very good revision.
- Coming back to my question: What is the reason to use two types of animals since both species healing occurs by wound shrinking? Your audience needs to understand it as well, which means that you must add this information in the text please.
- Have you tested if the compounds are allergen-free? Have you tested the endotoxin levels in your extracted compounds? These are assays that need to be run before you start any in vivo experiments mainly because you synthesised new compounds which have not characterised previously.
- l. 115 – 119; 131-134; 295-298; 325-329: Need English revision
- l. 136- 142: Please write as a scientific text not as a protocol and please revise English
- l. 229 – 230: “VEGF protein expressions increased…” should be “VEGF protein expression increased …” otherwise it makes no sense.
- l. 255: “…filtrate ointment but oral water only without Antrodia cinnamomea filtrate.” This has no sense
7. Fig. 5B: Please provide scale bars
Author Response
Response to Reviewer 1 Comments
Point 1: Coming back to my question: What is the reason to use two types of animals since both species healing occurs by wound shrinking? Your audience needs to understand it as well, which means that you must add this information in the text please.
Response 1: Thanks for point out this concern. We agreed reviewer’s comments. We had added the description in revised manuscript in method section 2.2.
Point 2: Have you tested if the compounds are allergen-free? Have you tested the endotoxin levels in your extracted compounds? These are assays that need to be run before you start any in vivo experiments mainly because you synthesised new compounds which have not characterised previously.
Rosponse 2:
Thanks for reviewer to point out this concern. Although we did not detect the endotoxin level of Antrodia cinnamomea extracts using commercial endotoxin analysis kit. However, we did perform experiments to verify the endotoxins level of AC extracts. In our unpublished data (attached as supplemental materials in this response), the AC extract was added into macrophage culture to test whether AC extract can activate macrophages. Because macrophage activation is considered to be correlated with the presence of endotoxins in assay, we have excluded the possible endotoxin contamination in our AC extract preparation based on the collected data. As you can see in attached data, when added AC extract into the RAW264.7 cell culture, i.e. a mouse macrophage cell line, no nitrate production was observed compared to the RAW264.7 cells stimulated with LPS or LPS plus interferon-g (supplement Fig. 1). Simultaneously, the RAW264.7 cell proliferation did not significantly alter after co-treatment with various doses of AC extracts, which suggests no cytotoxic effect of AC extracts on RAW264.7 cells (supplement Fig. 2).
.
Supplement Figure 1. Addition of the AC extract alone did not activate macrophages.
5x104 RAW264.7 cells were treated with medium alone, 0.1% DMSO/0.001% ethanol (vehicle), LPS (0.1mg/mL), LPS (0.1mg/mL) plus IFN-g (5U), or various doses of AC extracts (50, 100, 200, 400, 800 mg/mL) as indicated. Culture supernatants were collected after 24 h. The amounts of nitrite were measured by Griess assay. Data are expressed as mean of nitrite (mM) ± SD of four independent experiments.
Supplement Figure 2. The effect of AC extracts on the proliferation of RAW 264.7 cells.
5x104 RAW264.7 cells were treated with 0.1% DMSO/0.001% ethanol (vehicle) or various doses of AC extracts (50, 100, 200, 400, 800 mg/mL) as indicated. After 24 h incubation, the proliferation of RAW264.7 cells was measured by MTT assay. Data are expressed as mean of OD570nm ± SD of four independent experiments.
Point 3:l. 115 – 119; 131-134; 295-298; 325-329: Need English revision
Response 3: Thanks for reviewer’s comment. We have corrected.
Point 4:l. 136- 142: Please write as a scientific text not as a protocol and please revise English
Response 4: Thanks for reviewer’s comment. We have corrected.
Point 5: l. 229 – 230: “VEGF protein expressions increased…” should be “VEGF protein expression increased …” otherwise it makes no sense.
Response 5: Thanks for reviewer’s comment. We have corrected.
Point 6: l. 255: “…filtrate ointment but oral water only without Antrodia cinnamomea filtrate.” This has no sense
Response 6: Thanks for reviewer’s comment. We have corrected.
Point 7: Fig. 5B: Please provide scale bars
Response 7: We have added scale bar on figure 5B.

Reviewer 2 Report
The authors improved substantially the quality of the manuscript.
Author Response
Thank you very much.
Reviewer 3 Report
This revision has significantly improved the manuscript. The author addressed all my comments. I have nothing to add.
Author Response
Thank you very much.